# Comparing DNA Isolation and Preparation Protocols for Dried Blood Spots in the Context of Genomic Newborn Screening

**DOI:** 10.3390/ijns11030075

**Published:** 2025-09-03

**Authors:** Annelotte J. Duintjer, Sandra Imholz, Ingrid Pico-Knijnenburg, Adinda Heuperman, Hennie Hodemaekers, Eva S. Deutekom, Els Voorhoeve, Martijn E. T. Dollé, Mirjam van der Burg

**Affiliations:** 1Department of Pediatrics, Laboratory for Pediatric Immunology, Willem-Alexander Children’s Hospital, Leiden University Medical Center, 2333 ZA Leiden, The Netherlands; a.j.duintjer@lumc.nl (A.J.D.); i.pico-knijnenburg@lumc.nl (I.P.-K.); a.heuperman@lumc.nl (A.H.); 2Centre for Health Protection, National Institute for Public Health and the Environment, 3721 MA Bilthoven, The Netherlands; sandra.imholz@rivm.nl (S.I.); hennie.hodemaekers@rivm.nl (H.H.); eva.deutekom@rivm.nl (E.S.D.); els.voorhoeve@rivm.nl (E.V.); martijn.dolle@rivm.nl (M.E.T.D.)

**Keywords:** DNA isolation, DNA preparation, dried blood spot, DBS, newborn screening, NBS, genomic newborn screening, next-generation sequencing, NGS

## Abstract

Due to rapid technical advancements and increasing cost-effectiveness, the potential application of next-generation sequencing (NGS) in newborn screening (NBS) has raised great interest worldwide. Genomic NBS offers the possibility to improve current NBS programs when applied as follow-up tier, and, as first-tier, allows for inclusion of conditions lacking a detectable biomarker for conventional NBS. Obtaining enough high-quality DNA from typically limited dried blood spot (DBS) material to meet NGS requirements can be challenging. Selecting a DNA isolation method for genomic NBS requires balancing technical performance and laboratory feasibility with optimal cost-effectiveness. Ten DNA isolation protocols, including two column-based, five lysis-based, and three semi-automated magnetic bead-based protocols, were evaluated on technical outcomes and performance in targeted amplicon sequencing. Additionally, estimated costs, hands-on time, turnaround time, scalability, and plastic footprint were assessed. Although technical outcomes, including yield, purity, and molecular weight, differed between methods, qualitative results in amplicon sequencing, as defined by read output, mapping, and coverage depth, were found sufficient and comparable for various protocols. In conclusion, both technical requirements and operational parameters are crucial when selecting a DNA isolation protocol and will depend on the NGS application as well as the NBS approach, as either first-tier or follow-up tier.

## 1. Introduction

Newborn screening (NBS) programs are continuously expanding worldwide, resulting in new techniques and assays that are being incorporated in routine practice in screening laboratories [1]. A DNA-focused analysis was first introduced in NBS as a first-tier for severe combined immunodeficiency (SCID) by quantifying T-cell receptor excision circles (TRECs), circular DNA fragments excised during T-cell receptor rearrangement in the thymus [2,3]. Now, multiplex TREC kits have become available to additionally detect spinal muscular atrophy (SMA) and (X-linked) agammaglobulinemia (XLA) in a combined assay, thereby further expanding the application of DNA-based screening techniques on dried blood spots (DBSs) from NBS cards [4,5,6].

The rapid technical development and increasing cost-effectiveness of next-generation sequencing (NGS) have raised global interest in the potential role of NGS in NBS, thereby extending the application of DNA analysis in NBS [7,8]. NGS could expand NBS programs with genetic diseases that significantly benefit from early detection and meet the Wilson and Jungner criteria, but lack a specific biochemical marker to be detected via conventional NBS techniques [9,10]. Multiple international pilot studies exploring NGS in NBS are currently being, or have been, conducted both as first-tier and as second-tier applications and with different NGS approaches, including targeted sequencing, whole exome sequencing (WES), and whole genome sequencing (WGS) [11,12,13,14,15,16,17,18,19].

Dependent on the NGS approach, certain qualitative and quantitative requirements apply to the DNA input to yield optimal sequence results. However, isolating DNA from often limited available DBS material that meets the requirements for further NGS applications can be challenging. In addition to ensuring optimal quality and quantity, operational parameters of the DNA isolation method regarding costs, hands-on time, turnaround time, scalability, and sustainability are of utmost importance in the context of NBS, where often large numbers of samples need to be analyzed per day while maintaining cost-effectiveness.

We evaluated six DNA isolation methods for DBSs within a total of ten protocols, including two column-based protocols, five lysis-based protocols, and three semi-automated protocols based on magnetic bead capture, and compared qualitative and quantitative outcome measurements. Furthermore, an operational evaluation was performed to assess costs, hands-on time, turnaround time, scalability, and plastic footprint. Finally, the performance of the DNA isolation protocols was assessed for a targeted amplicon sequencing application.

## 2. Materials and Methods

### 2.1. Samples

Peripheral blood samples from seven anonymous adult donors were obtained from the Leiden University Medical Center (LUMC) Healthy Voluntary Donor Service (LuVDS; application number 24.025). This sample size was approximated to be sufficient for providing a robust comparison of the DNA isolation methods, while remaining feasible within downstream experimental constraints. Subsequently, blood collected in EDTA tubes was dripped on filter cards (Whatman 903TM Specimen Collection Paper) to create DBSs. Each DNA isolation protocol was applied to one, two, and three 3.2 mm DBS punches per donor, punched from random DBS sites within the filter cards. This study was subject to the non-Medical Research Involving Human Subjects act (nWMO) (nr. 23-3127) and was therefore exempt from review by the Medical Ethics Review Committee.

### 2.2. DNA Isolation Protocols

Six DNA isolation methods were evaluated, with four protocol variations for a lysis-based method and two protocol variations for a magnetic bead-based method. This resulted in a total of ten DNA isolation protocols tested, including two manual column-based protocols, five manual lysis-based protocols, and three semi-automated protocols based on capture by magnetic beads. For protocol 1 (‘Adapted Sigma Mini’), DNA was isolated with the GenElute^tm^ Mammalian Genomic DNA Miniprep Kit (Sigma-Aldrich, St. Louis, MO, USA) according to the protocol for isolating genomic DNA from DBSs from the QIAamp DNA Micro^®^ Handbook (second edition May 2010) (QIAGEN, Hilden, Germany) with a final elution volume of 20 µL. In protocol 2 (‘QIAamp Micro’), the QIAamp DNA Micro^®^ kit (QIAGEN, Hilden, Germany) and its corresponding protocol were used with a final elution volume of 20 µL. With protocol 3 (‘QIAGEN ES-1’), the QIAGEN Generation DNA Elution Solution 2 (QIAGEN, Hilden, Germany) was used according to the two-step protocol described by Strand, et al. (2022) [12], which consisted of a washing step with 150 µL elution solution at 60 °C for 15 min at 500 rpm followed by eluting in 100 µL of the same elution solution at 99 °C for 30 min at 500 rpm. Protocol 4 (‘QIAGEN ES-2’) was the same as protocol 3, but with a shorter elution step of 15 min. In protocol 5 (‘QIAGEN ES-3’), the same elution solution was used as in protocols 3 and 4, but with the two-step protocol from Hendrix, et al. (2020) [20], where washing was performed with 150 µL elution solution at room temperature for 15 min at 500 rpm followed by elution in 50 µL at 99 °C for 30 min at 500 rpm. Protocol 6 (‘QIAGEN ES-4’) was a variation on protocol 5 with a shorter 15 min elution step. For protocol 7 (‘Thermo Fisher’), the Applied Biosystems^tm^ DNA Extract All Reagents Kit (Thermo Fisher Scientific, Waltham, MA, USA) and an adapted protocol for blood card samples were used. The adopted protocol consisted of an additional first washing step by adding 150 µL washing buffer (0.5 g Thesit and 10 mL PBS 1× pH 7.4), followed by centrifuging for 1 min at 2400 rpm and then vortexing for 5 min at 1500 rpm. Afterwards, the supernatant was discarded and a second wash step was performed with 150 µL nuclease-free water. Then, 5 µL all reagent lysis was added, followed by incubation at 95 °C for five minutes. After returning to room temperature, DNA was eluted in 35 µL stabilizing buffer. In protocol 8 (‘Maxwell’), DNA isolation was performed on the Maxwell^®^ RSC 48 instrument using the Maxwell^®^ RSC FFPE Plus DNA Kit (Promega, Madison, WI, USA) with its corresponding protocol for automated DNA purification from blood on FTA cards. Protocol 9 (‘Chemagic Overnight’) used the Chemagic^tm^ DNA Blood Spot 13 Kit 96H (Revvity, Waltham, MA, USA) and its corresponding protocol, without using DTT, on the Chemagic^tm^ 360 instrument. Samples were incubated at 56 °C overnight for sixteen hours at 800 rpm agitation. DNA was eluted in 55 µL. Finally, protocol 10 (‘Chemagic Short’) was identical to method 9, but with a short incubation at 56 °C for three hours.

### 2.3. Comparative Measurements

#### 2.3.1. Practical Outcomes

Hands-on time and turnaround time were documented during the performance of the DNA isolation procedures. The turnaround time was estimated excluding punching the DBSs, quantification of DNA, or further measurements. Sustainability and scalability were assessed in relation to the other tested DNA isolation protocols. Conclusions regarding sustainability and the scalability of the protocol in a screening laboratory were based on individual experiences in the research group that were logged after each DNA isolation procedure. These parameters were displayed based on a five-point Likert scale (poor–fair –average–good–excellent). Sustainability was presented as the plastic footprint by estimating the amount of laboratory consumables that were required to complete the protocols. Scalability was based on turnaround capacity to scale up to 96 samples per experiment. Finally, costs per sample were determined based on publicly available information on websites or in documents of the corresponding manufacturer without discounts. The estimated costs were based on the kits and buffers used and did not include costs for instruments and other consumables.

#### 2.3.2. Technical Outcomes

DNA concentration (ng/µL) was measured on the Invitrogen Qubit^tm^ Fluorometer (Thermo Fisher Scientific, Waltham, MA, USA) using 1 µL of sample in 199 µL Qubit^tm^ 1X dsDNA High Sensitivity working solution. DNA purity was assessed on the NanoDrop Spectrophotometer (Thermo Fisher Scientific, Waltham, MA, USA) using 1.5 µL of sample. For protocol 7 (‘Thermo Fisher’), 1.5 µL of sample was measured in a 1:10 dilution with nuclease-free water due to negative OD ratio results when added undiluted. A 260/280 ratio of 1.8 is generally considered optimal for pure DNA. A range within 10% of this value (1.62–1.98) was considered optimal in our analyses. For the A260/A230 ratio, values between 2.0 and 2.2 were regarded to be most optimal.

For measuring molecular weight, seven of the ten protocols were selected. The protocols included the five protocols, with both Chemagic protocol variants: protocol 1 (‘Adapted Sigma Mini’), protocol 2 (‘QIAamp Micro’), protocol 5 (‘QIAGEN ES-3’), protocol 7 (‘Thermo Fisher’), protocol 8 (‘Maxwell’), protocol 9 (‘Chemagic Overnight’), and protocol 10 (‘Chemagic Short’). Protocol 5 was selected over protocols 3, 4, and 6 due to comparable performance and lower elution volume (results 3.4). The molecular weight was measured in a diluted concentration of 0.3 ng/µL on the Femto Pulse System (Agilent Technologies, Santa Clara, CA, USA) using the Genomic DNA Analysis kit (Agilent Technologies, Santa Clara, CA, USA) with a lambda ladder (Bio-Rad Laboratories, Hercules, CA, USA). Samples were stored in the freezer at a temperature of −20 °C between measurements.

#### 2.3.3. Sequencing

The seven protocols included in the molecular weight analysis were also compared based on targeted amplicon sequencing performance. Sequencing was performed on DNA from three different donors per DNA isolation protocol, and only DNA isolated from one and two DBS punches (3.2 mm) was used to reflect the limited DBS material typically available in NBS. As a control, a genome in a bottle (GIAB) reference DNA sample (NA12878/HG001) was included. Libraries were prepared following the AmpliSeq^tm^ for Illumina protocol for two primer pools with a custom panel consisting of 3003 amplicons covering 105 genes associated with a phenotype of low TRECs and T-cell lymphopenia at birth (Illumina, San Diego, CA, USA). When available, 20 ng of DNA was included as input material in accordance with the protocol, or, alternatively, the maximum input possible was used (minimum of 3 ng). Regardless of the DNA input amount, targets were amplified by 15 PCR cycles. Libraries were quantified using SYBR Green-based quantitative PCR (qPCR) (Kapa Biosystems, Wilmington, MA, USA) and diluted to a starting concentration of 0.1 nM prior to pooling. Subsequently, the library pool was spiked in with 5% PhiX Control V3 (Illumina, San Diego, CA, USA). Sequencing was performed on the NextSeq550^tm^ system with the NextSeq500/550 Mid Output Kit v2.5 (300 cycles; Illumina, San Diego, CA, USA). BCL files were demultiplexed and converted to FASTQ files by an in-house pipeline using Illumina bcl2fastq2 Conversion Software v2.20.0.422 (Linux rpm). Secondary data analysis was performed using the Illumina DRAGEN pipeline (4-2-4-v2). Outcomes including sequencing yield defined by the number of output reads and other quality parameters generated from the DRAGEN pipeline were assessed. For mapping performance, the percentage of unmapped reads was determined. The mean read depth over the target region was calculated from the number of uniquely mapped bases to the target region divided by the number of amplicons in the target region (*N* = 3003). To assess coverage over the target region, the percentage of amplicons with a read depth of more than 30x was calculated.

### 2.4. Statistical Analysis

Statistical analysis was performed in IBM SPSS Statistics (version 29.0.0.0 (241)). A Kruskal–Wallis test was used to assess statistically significant differences in the DNA yield between the isolation protocols. Upon finding a significant result, a post-hoc pairwise comparison was performed to identify which protocols differed using Dunn’s test with Bonferroni correction to adjust for multiple testing. A two-sided *p*-value of <0.05 was considered statistically significant.

The dispersion (or variability) of DNA yield around the mean was assessed using the coefficient of variation (CV). The CV is a standardized measure of how much the DNA yield fluctuates between different samples for each protocol. The CV was defined as the ratio between the standard deviation and the mean DNA yield. A lower CV indicates that the method produces more consistent results, while a higher CV means there is more variation in the yield.

The hypothetical linear function assuming a direct proportional increase in DNA yield when increasing DBS punches as input material was calculated based on the yield from one DBS punch. The means and 95% confidence intervals (CIs) for the means were calculated and plotted in the same graph for comparison. For visualization of data, GraphPad Prism version 10.2.3 (403) was used.

## 3. Results

Ten different DNA isolation protocols were performed on DBSs from a total of seven healthy donors (Table 1). For each protocol, one, two, and three DBS punches (3.2 mm) were tested per individual donor to compare DNA yield for an increasing amount of input material. Hence, in total, 210 DNA isolations were performed.

### 3.1. Operational Performance

Operational performance is crucial for an efficient and cost-effective NBS program; therefore, practical parameters were evaluated for all ten DNA isolation protocols, including costs, hands-on time, turnaround time, scalability, and plastic footprint (Table 1). The costs were lowest for the lysis-based methods, particularly the two-step protocols 3, 4, 5, and 6 that used the QIAGEN Generation DNA Elution Solution 2. The highest costs were estimated for the semi-automated protocol 8 (‘Maxwell’).

Protocols 3, 4, 5, and 6 using the QIAGEN DNA Elution solution 2 also had the shortest turnaround time of 45 min for protocols 4 and 6 and one hour for protocols 3 and 5. Moreover, these protocols were considered most scalable as they only require two steps for DNA isolation. Due to its short two-step protocol, which requires fewer consumables as compared with the other protocols, these protocols were also superior based on plastic footprint. Although the magnetic bead-based methods use an automated instrument, protocol 8 (‘Maxwell’) was ranked inferior based on scalability compared to protocols 9 (‘Chemagic Overnight’) and 10 (‘Chemagic Short’), as protocol 8 requires a manual transfer of DBS punches into a new tube. The magnetic bead-based methods, as well as the column-based methods, performed worse based on plastic footprint as compared to the lysis-based methods due to the requirement of multiple plastic tubes per sample and, specifically for the Chemagic methods, multiple plates.

**Table 1 IJNS-11-00075-t001:** DNA isolation protocols and operational parameters.

Number	DNA Isolation Protocols	Technique	Total Elution Volume (µL)	Costs (€)	Hands-on Time (Hours)	Turnaround Time (Hours)	Scalability	Plastic Footprint
1	Adapted Sigma Mini	Column, manual	20	2–4	1.5	2.75	Poor	Fair
2	QIAamp Micro	Column, manual	20	5–7	1.5	2.75	Poor	Fair
3	QIAGEN ES-1 (30′–100 μL elution)	Lysis, manual	100	<1	0.25	1	Excellent	Excellent
4	QIAGEN ES-2 (15′–100 μL elution)	Lysis, manual	100	<1	0.25	0.75	Excellent	Excellent
5	QIAGEN ES-3 (30′–50 μL elution)	Lysis, manual	50	<1	0.25	1	Excellent	Excellent
6	QIAGEN ES-4 (15′–50 μL elution)	Lysis, manual	50	<1	0.25	0.75	Excellent	Excellent
7	Thermo Fisher	Lysis, manual	40	1–2	0.5	1	Average	Average
8	Maxwell	Magnetic beads, semi-automated	50	8–10	0.5	1.5	Fair	Fair
9	Chemagic Overnight	Magnetic beads, semi-automated	55	1–2	1	18	Good	Poor
10	Chemagic Short	Magnetic beads, semi-automated	55	1–2	1	5	Good	Poor

Costs are estimated within ranges and are indicated per sample. Values for scalability (defined as the possibility to scale up to 96 samples per experiment) and plastic footprint (defined as the amount of plastic waste) are presented according to a five-point Likert scale.

### 3.2. DNA Yield

The median and ranges of the yield, concentration, and OD ratios are depicted in Figure 1 and Appendix A. An overview of all means and standard deviations is included in Appendix A. For protocol 4 (‘QIAGEN ES-2’), one donor sample with one DBS punch was excluded due to failure to yield DNA despite multiple extractions.

Figure 1A shows the DNA yield per DNA isolation protocol, as calculated from the concentration multiplied by the elution volume. For all DNA isolation protocols, an increase in median DNA yield was seen for increasing amounts of DBS punches used as starting material, indicating that three DBS punches resulted in the highest median DNA yield. DNA yield differed significantly between the DNA isolation protocols across all DBS punches (Kruskal–Wallis, *p* < 0.001). The post-hoc pairwise comparisons are shown in Appendix A. The magnetic bead-based protocols 8 (‘Maxwell’), 9 (‘Chemagic Overnight’), and 10 (‘Chemagic Short’) showed the highest median DNA yield of 35.90 ng, 65.45 ng, and 89.10 ng, respectively, for one DBS punch; 148 ng, 214.50 ng, and 189.20 ng for two DBS punches; and 328 ng, 368.5 ng, and 256.30 ng for three DBS punches (Figure 1A; Appendix A) compared to the other methods. Because the magnetic-based protocols showed the highest yields, we wanted to assess whether the magnetic-based protocols differed from each other in yield. However, no significant differences in DNA yield were observed between the magnetic bead-based methods (Dunn’s test with Bonferroni correction, all *p* = 1.000) (Appendix A).

It is also important to assess how consistent the DNA yields are. For practical applications, a method that produces stable and predictable results is often preferred over highly variable ones. Therefore, we additionally evaluated the dispersion of the DNA yields for each method, as measured by CV (see Section 2.4). The CV did not significantly differ between the ten methods, showing no difference in the stability of the results between methods.

Ideally, a DNA isolation protocol should enable a linear increase in DNA yield with increasing input material, such as the number of DBS punches, indicating that the extraction system is not saturated and scales proportionally within the tested range. In theory, this means that doubling the number of DBS punches results in a twofold increase in DNA yield. For protocol 7 (‘Thermo Fisher’), the observed increase was less than the assumed direct proportionality, with the hypothetical values 49.80 ng for two DBS and 74.70 ng for three DBS punches lying above the observed 95% CI (26.81–37.46 ng and 30.79–44.03 ng, respectively) (Figure 2; Appendix A). This sublinear increase is indicative of the saturation and suboptimal performance of the DNA isolation protocols with an increasing number of punches. For the DNA yield from three DBS punches with protocol 5 (‘QIAGEN ES-3’), this was also observed, where the yield under direct proportional increase hypothetically would be 61.98 ng (95% CI: 27.91–47.69 ng). For protocol 8 ‘Maxwell’, on the contrary, the hypothetical direct proportional yield for three DBS punches of 141.03 ng was below the observed 95% CI (286.80–405.20 ng), implying optimal performance when isolating from three DBS punches.

### 3.3. DNA Purity

The nucleic acid purity defined by the A260/A280 ratio is shown in Figure 1B. For protocol 8 (‘Maxwell’), the median A260/A280 ratios for all numbers of DBS used fell within the 10% range from the optimal ratio of 1.8 (see Section 2.3.2). Other median A260/A280 ratios that fell within the 10% range from the optimal ratio were the DNA isolated from two and three DBS punches with protocol 1 (‘Adapted Sigma Mini’), three DBS punches with protocol 2 (‘QIAamp Micro’), and three DBS punches from protocol 9 (‘Chemagic Overnight’). For other protocols, there was no median A260/A280 ratio that fell within the 10% range from 1.8. For the median A260/A230 ratio, none of the DNA isolation protocols fell within the optimal range of 2.0–2.2 (Appendix A). The low 260/280 and 260/230 ratios may be hampered by low DNA concentrations outside the linear range of the NanoDrop Spectrophotometer (Appendix A).

### 3.4. Molecular Weight

The molecular weight of the isolated DNA was compared between seven protocols by assessment on the Femto Pulse system. From the four QIAGEN elution solution protocols, we selected only QIAGEN ES-3 for further analysis because all four protocols had similar technical parameters, and a lower elution volume was preferred for the subsequent analyses. Figure 3 shows the analysis results for two different donors per protocol with DNA isolated from one DBS punch. The corresponding electropherograms are presented in Appendix A. Comparable high molecular weight was observed between magnetic bead-based protocols 9 (‘Chemagic Overnight’) and 10 (‘Chemagic Short’), with modes around 48–60 kb and a relatively narrow range. The other magnetic bead-based protocol 8 (‘Maxwell’) had lower modes of around 2–6 kb, and low intensity in one donor. The column-based protocols 1 (‘Adapted Sigma Mini’) and 2 (‘QIAamp Micro’) showed a comparable broad range with modes of around 16–22 kb and 11 kb, respectively. DNA isolated with protocol 5 (‘QIAGEN ES-3’) showed a smear with low intensity. In Appendix A, the results are shown for DNA isolated from two and three DBS punches for the same donors as presented in Figure 3. Within protocols, no notable differences in molecular weight were observed between donors and between different amounts of DBS punches used as starting material, except for protocol 7 (‘Thermo Fisher’), which showed differences between all measurements.

### 3.5. Sequencing Performance

Seven different DNA isolation protocols on three donors were evaluated based on performance in targeted amplicon sequencing. Library preparation was performed for a total of 42 DNA samples isolated from one DBS punch and two DBS punches. Two libraries were excluded due to preparation failure, one with DNA isolated from two DBS punches using protocol 1 (‘Adapted Sigma Mini’) and one with DNA isolated from two DBS punches with protocol 2 (‘QIAamp Micro’), leaving 40 libraries for sequencing. Together with the GIAB control sample and two internal controls, a total of 43 libraries were included on one flow cell.

The sequence run yielded an overall output of 53.3 Gb. Other primary analysis metrics included a cluster density of 226.2 k/mm^2^, a cluster passing filter of 86.9%, and a Q30 of 82.2%. Three samples in which DNA was isolated with protocol 2 (‘QIAamp Micro’) were excluded from further analyses, as these libraries had a molarity below the applied dilution cut-off of 0.1 nM, but were added nevertheless and eventually had no sequence yield. Two out of the three excluded samples were from the same donor, with DNA isolated from both one DBS punch and two DBS punches. The third sample came from a different donor with DNA isolated from one DBS punch.

In all analyzed samples (*N* = 37), sequencing generated a minimum of 2.98 million reads and resulted in a read depth over the target region of at least 220× (Figure 4A,C). A relatively high median percentage of unmapped reads of 34.07% for one DBS punch and 33.75% for two DBS punches was seen with protocol 7 (‘Thermo Fisher’) (Figure 4B). For other protocols, more comparable percentages of unmapped reads were observed with medians ranging from 8.15% to 15.56%. The median coverage, as defined by the percentage of amplicons (*N* total = 3003) with >30× read depth, was for all protocols at least 99%. Protocol 7 (‘Thermo Fisher’) showed evident outliers for coverage in two samples of different donors, one with DNA isolated from one DBS punch (97.20%) and one from two DBS punches (88.58%). There were five amplicons (0.17%) with less than 30× read depth in all samples, regardless of DNA isolation protocol or donor.

Thus, in five of the seven protocols, regardless of the number of DBSs initially used for DNA isolation, amplicon sequencing resulted in a high yield and mean read depth over the target region, with also high and comparable median coverage. It should be noted that one sample (16.7%) failed for protocol 1 (‘Adapted Sigma Mini’) and four samples (66.7%) failed for protocol 2 (‘QIAamp Micro’) due to insufficient library yield.

**Figure 4 IJNS-11-00075-f004:**
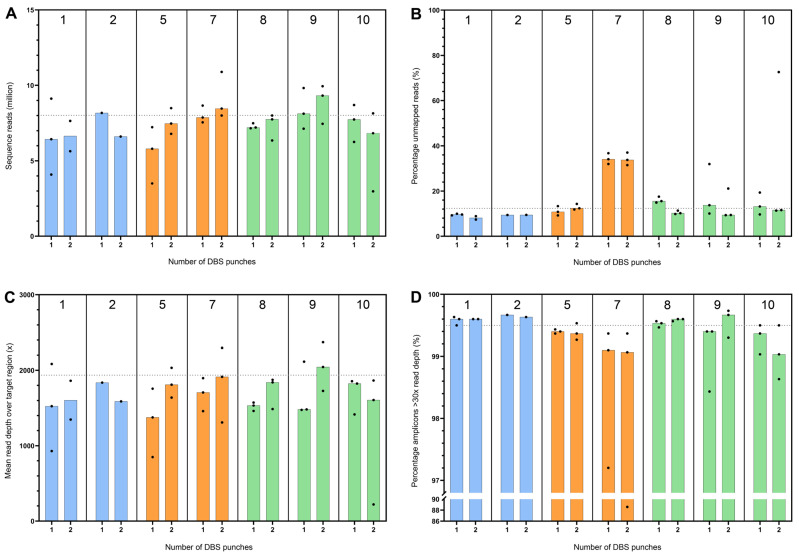
Sequencing performance based on yield, mapping, and coverage. DBS = dried blood spot. Comparison of amplicon sequencing performance between seven different DNA isolation protocols applied on one and two DBS punches (3.2 mm) of starting material, including the generated sequence reads in millions (**A**), percentage unmapped reads (**B**), mean read depth over target region (**C**) and percentage of amplicons (*N* total = 3003) with >30× read depth (**D**). The numbers above the columns correspond to the DNA isolation protocols described in Table 1. The bar reaches the median from all samples per protocol, with colors indicating different techniques used, including column-based (blue), lysis-based (orange), and magnetic bead-based (green). The dotted line depicts outcomes for the GIAB control sample.

## 4. Discussion

DNA extraction from DBSs is becoming increasingly relevant in NBS due to the growing interest in NGS applications in screening laboratories [21,22]. Globally, multiple studies are exploring the potential role of NGS in NBS, either as a first-tier approach to expand NBS programs with eligible genetic disorders that lack suitable biomarkers for current conventional techniques or as a second-tier approach to optimize ongoing NBS programs by reducing non-actionable secondary findings and false positives [11,12,13,14,15,16,17,18,19].

Prior to integrating a DNA extraction protocol into a NGS workflow in NBS, it is essential to carefully select the protocol that meets the technical requirements of the intended NGS application and can also be realistically used in a screening laboratory, considering factors such as cost-effectiveness and turnaround time. In this study, a comprehensive evaluation of ten different DNA isolation protocols for DBSs was carried out based on different technical and operational performance as well as qualitative outcomes in targeted amplicon sequencing.

Notable differences were observed in technical outcomes between the evaluated DNA isolation protocols, which included two column-based protocols, five lysis-based protocols, and three semi-automated protocols using magnetic beads. Depending on the NGS approach, certain requirements of the DNA become critical to reach high-quality sequence results. Although the seven protocols included for sequencing differed based on technical outcomes after DNA isolation, comparable amplicon sequence performance was observed in five of the seven protocols, as evidenced by a high number of generated reads, high mean read depth over the target region, and high median coverage as defined by the percentage of amplicons with >30× read depth. However, it should be noted that samples were excluded for protocols 1 (‘Adapted Sigma Mini’) and 2 (‘QIAamp Micro’) due to insufficient library yield. In addition, method 7 (‘Thermo Fisher’) resulted in a consistent higher percentage of unmapped reads than the other protocols tested. For the remaining protocols 5, 8, 9, and 10, no notable differences in sequence performance were observed between DNA isolated from one DBS punch and from two DBS punches (3.2 mm). The application of amplicon sequencing may not demand high DNA quantity, purity, and molecular weight, but it does lead to failures for some DNA isolation protocols. Moreover, protocol 4 (‘QIAGEN ES-2), although not tested for sequencing performance, failed to produce sufficient DNA from one DBS punch input from a single donor repeatedly. Therefore, protocols 1, 2, 4, and 7 should undergo further optimization before being applied in a NBS setting, even when amplicon sequencing is used and especially when only one DBS punch is available as starting material. For the other procedures 5, 8, 9, and 10, the selection of a suitable DNA isolation protocol for amplicon sequencing could therefore be more strongly based on operational parameters focusing on minimizing costs, reducing hands-on and turnaround time, and minimizing the plastic footprint.

When extending the sequenced genomic region by either performing WES, WGS, or long-read NGS, technical requirements can become more important, and some protocols might be less suitable due to lower DNA yield, purity, and molecular weight [23]. For all ten protocols, increasing input material to a maximum of three DBS punches (3.2 mm) resulted in an increase in DNA yield. When requiring a large input of high-quality DNA, the semi-automated magnetic beads protocols performed superior based on DNA yield, and for protocol 8 (‘Maxwell’), also for the A260/A280 ratio compared to the other protocols. Specifically, the methods using the Chemagic^tm^ 360 instrument also yielded high molecular weight. It is expected that DNA isolation from neonatal DBS samples will result in similar or even higher DNA yield, also after storage time in biobanks, because the number of leukocytes per blood volume is higher [24,25]. However, when handling stored NBS samples, one should consider assessing the fragmentation of DNA, which could occur with an increasing period of storage [24].

Current projects on NGS in NBS focus on short-read sequencing approaches. As costs are expected to further decrease and accuracy to increase, long-read sequencing in NBS will likely be explored more extensively. This technology could potentially overcome mapping issues and incomplete coverage in genes with homologous regions, such as paralogs, pseudogenes, or other regions of high sequence similarity (e.g., repeats), and improve the detection of structural variants [26,27,28,29,30]. However, the technical feasibility of using DNA isolated from DBSs in long-read sequencing applications faces significant challenges related to DNA quantity, quality, and molecular weight [31,32]. Nonetheless, from a DNA-extraction point of view, it appears feasible provided that sufficient DBS input is available and one of the automated magnetic bead-based protocols is used.

Besides the NGS application, selecting the most suitable DNA isolation protocol also depends on the screening approach. For follow-up tiers such as second-tier NGS, small numbers are generally expected, indicating that a DNA isolation protocol does not need to be scalable due to low throughput. In that case, manual protocols might be preferred over semi-automated protocols as these are generally lower in costs, require less processing time, and use fewer consumables, making them more sustainable. In contrast to second-tier applications, first-tier NGS will handle significantly higher throughput. For first-tier approaches, a DNA isolation protocol should be selected that allows for large numbers of NBS samples to be analyzed daily within an acceptable time frame. A DNA isolation protocol that could handle high-throughput either requires automation or could potentially be manual, provided that the protocol is simple, short, and scalable. Of the protocols evaluated, these requirements are best suited by either the semi-automated methods using the Chemagic^tm^ 360 instrument or the short two-step protocol 5 (‘QIAGEN ES-3). However, the Chemagic^tm^ 360 instrument enables processing of a maximum of 96 samples per run, indicating that when the daily throughput is expected to be larger than 96, multiple runs are required to process all samples. Therefore, if the NGS approach is compatible with smaller quantities of DNA with lower purity and molecular weight, a short and cheaper manual lysis-based protocol could also be considered for first-tier applications.

The different evaluated DNA isolation protocols for DBSs all have their advantages and limitations based on technical outcomes and operational performance. When selecting an optimal protocol for NGS in NBS, one should carefully compare the requirements for the intended NBS application and the applied sequencing approach. Our findings, however, demonstrate that for this targeted amplicon sequencing approach, comparable results were found based on qualitative sequence parameters, allowing for prioritization of cost-effective and time-efficient protocols over purely optimal technical outcomes when selecting a suitable DNA isolation protocol. For other NGS applications, performance will, however, be more influenced by technical parameters of the DNA. Ultimately, balancing technical performance with laboratory feasibility while maintaining optimal cost-effectiveness will be crucial for integrating DNA sequencing into NBS programs.

## Figures and Tables

**Figure 1 IJNS-11-00075-f001:**
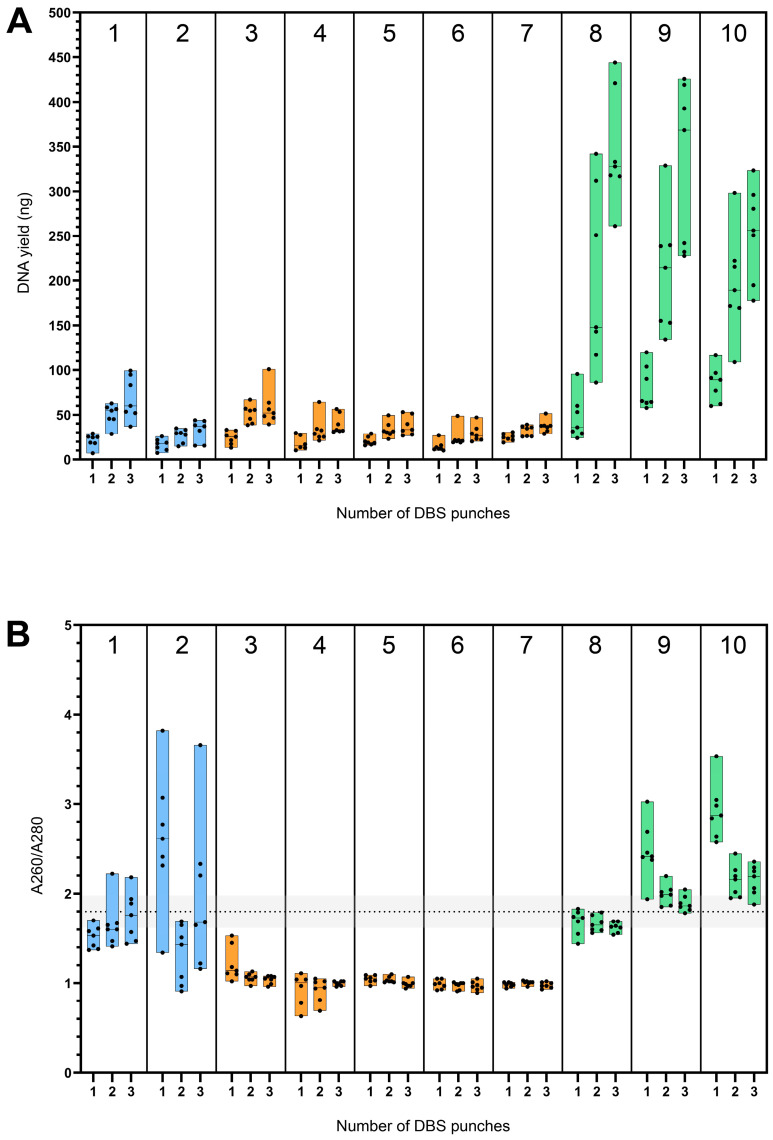
DNA yield and A260/A280 ratio. DBS = dried blood spot. Comparison of DNA yield (ng) (**A**) and A260/A280 (**B**) ratio between ten different DNA isolation protocols using one, two, and three DBS punches (3.2 mm). The dotted line depicts the optimal A260/A280 ratio (**B**) with a 10% interval around it in grey. The numbers above the columns correspond to the DNA isolation protocols as described in Table 1. The median and range for the measurements per protocol are depicted by the bars, with colors indicating different techniques used, including column-based (blue), lysis-based (orange), and magnetic bead-based (green).

**Figure 2 IJNS-11-00075-f002:**
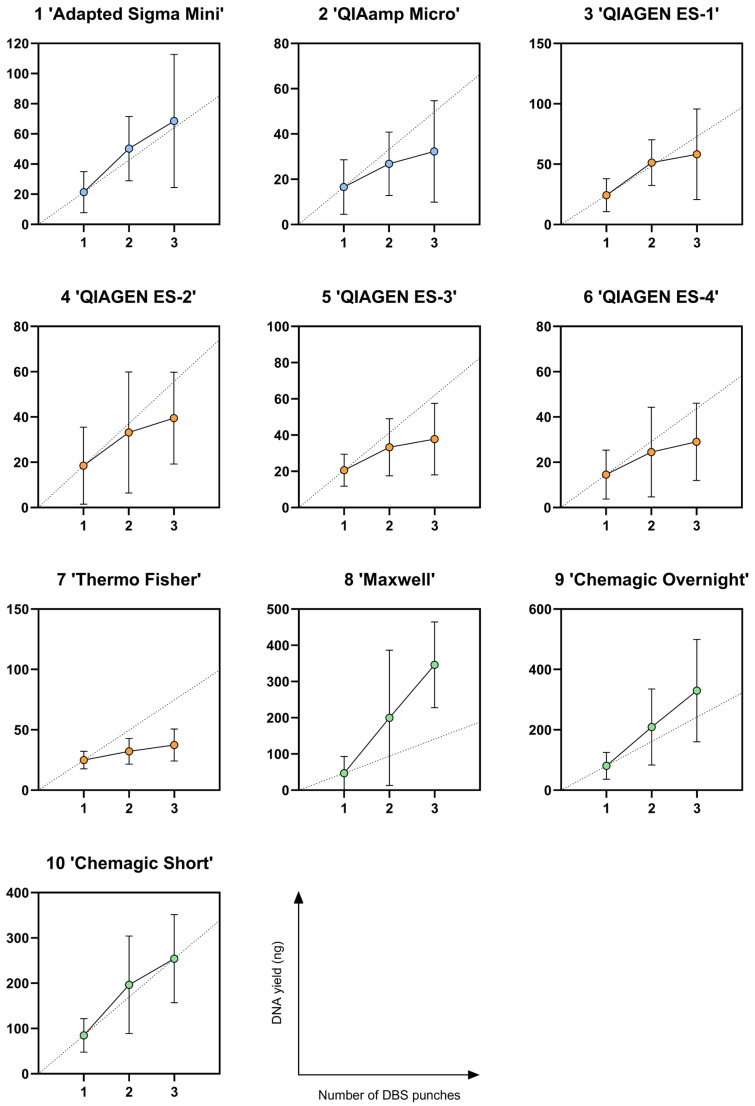
Mean DNA yield with increasing number of DBS punches. DBS = dried blood spot. Comparison of the mean DNA yield (ng) and 95% CI observed with increasing numbers of DBS punches (3.2 mm) used as starting material. The dotted line represents a hypothetical function if the yield from two and three DBS punches increased in direct proportion to the yield from one DBS punch. The colors indicate different techniques used, including column-based (blue), lysis-based (orange), and magnetic bead-based (green).

**Figure 3 IJNS-11-00075-f003:**
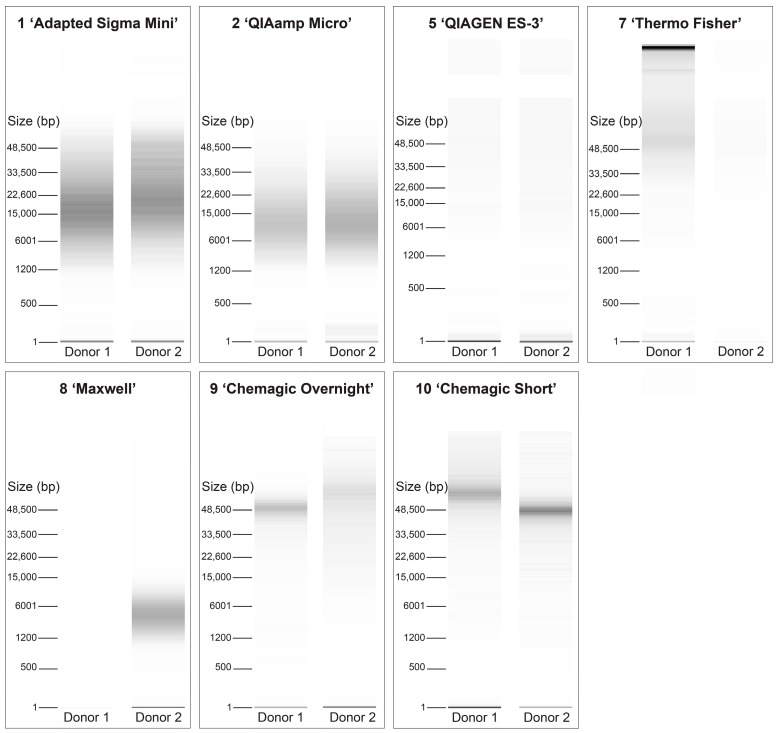
Molecular weight of isolated DNA from one DBS punch. DBS = dried blood spot. The analysis was performed with the Femto Pulse System on DNA isolated from one DBS punch (3.2 mm). For each DNA isolation protocol, results from two different donor samples are presented.

## Data Availability

All generated data in this study are presented in the main manuscript and in the Appendix A.

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
