# Peer review of "Comparing DNA Isolation and Preparation Protocols for Dried Blood Spots in the Context of Genomic Newborn Screening"

_2409-515X, 2025, doi:10.3390/ijns11030075_

Round 1
Reviewer 1 Report
Comments and Suggestions for Authors
My compliment to Authors for huge work done. I have just one comment or curiosity
Why such limited sample size with DBS were obtained from only 7 adult donors?
Can you clarify and write clearly on the paper the reason of your choice ?
Could you even explain why you did not analyze performances on neonatal DBS samples? As the target is newborn screening this could have been more representative and may be you would be able to detect differences in cellular content and DNA quality beteween methods. I would greateful if you could cite a comment on this issue in the final discussion.
Reviewer 2 Report
Comments and Suggestions for Authors
This manuscript presents a timely and relevant contribution to the ongoing international interest on newborn screening, particularly in the context of strategic planning and research investment. It is well composed, with clear and grammatically sound language throughout. Each section effectively delineates the study’s objectives, methodologies, findings, and corresponding discussion. The figures and tables are appropriately designed, accurately presented, and enhance the clarity and validity of the work. Some specific additional comments include: -
- Consent and Ethics Approval: Although no ethics is required for this study it would have been appropriate to state this at the start of the materials and methods section
- Samples - research design. Had the research group considered extracting DNA from the adult donors and comparing this to the same blood added to DBS filters. That would have been a good comparator against all the DBS extraction methods. A limitation of using adult blood is that in ‘real-life’ the testing will be carried out in the majority on newborn blood samples, but the authors have addressed this.
- DNA isolation methods - the research team have a selected a wide range of DNA extraction methods that are used routinely across genomic laboratories. This is great.
- Sequencing protocol and statistical analysis appropriate.
- Table 1: can the units be added to the total elution volume.
- Comparative measurements were grouped into operational performance, DNA yield, DNA purity, and sequencing performance. Whilst hands-on time and cost of plastics is important, ultimately, depth of coverage has the most significant impact on the ability to identify variants in NGS pipelines which are dependent on DNA quantity and quality.
- The research team have selected a targeted amplicon sequencing platform - it should be noted that PCR amplification steps can introduce coverage biases across the genome and even introduce incorrect bases during amplification. These limitations need to be considered. The authors are correct to point out in the discussion that the application of amplicon sequencing does not demand high DNA quantity, purity and molecular weight. But WES, WGS and LRS does. Interpretation in the discussion reflects the different limitations and challenges.
- It is not clear, and the authors do not explain why the library preparations were performed on one and 2 punches when 3 punches had resulted in the highest median DNA yield.
- It is surprising to see that the manual methods having less hands-on time and were more scalable than automated methods.
- It is interesting to note that the magnetic-based protocols showed the highest median DNA yield. It was also interesting to note that number of punches directly impacted the DNA yield as you would have anticipated. Sequencing technologies require a minimum amount of DNA to generate readable data. Low yield can lead to failed runs or incomplete data. Higher yield allows for deeper sequencing coverage, which improves the detection of rare variants and reduces errors. I feel that the authors addressed this adequately in the paper.
- The sequencing performance had the expected results, considering the number of punches used, and that data was based on a targeted amplicon sequencing.
Reviewer 3 Report
Comments and Suggestions for Authors
- In my understanding of the manuscript, the authors, through the comparison of ten DNA isolation protocols from DBS, and the evaluation of different technical and operational parameters of the various protocols, provide information that could be helpful to neonatal screening programs in making decisions about the usefulness and applicability of any of these protocols to their newborn screening programs that include or could potentially apply next generation sequencing (NGS) in screening and/or follow-up.
- I believe the topic is relevant to the field, although not entirely original because other research groups have already evaluated DNA isolation protocols with the same purpose (in the publication they reference some of these studies). However, it is interesting to highlight that in this manuscript, the authors evaluate certain modified protocols starting from established commercial methods, which could serve as a reference for future studies aiming to improve current DNA isolation protocols.
- My consideration is that the application of NGS to NBS programs is an emerging and challenging topic, and any new information, if it is obtained in a rigorously scientific manner, can fill a gap in knowledge or technical/operational applicability.
- The manuscript takes a rather comprehensive approach to the applicability of these DNA isolation protocols in technical and operational terms, discussing and suggesting, based on the current knowledge in the field, which protocols might be the most suitable depending on the requirements of the laboratory and/or the NBS program.
- This reviewer did not point out any issues with the methodology used by the authors. In general terms, the authors described in detail the protocols that were going to be evaluated, the parameters they included in that technical and operational assessment, the sequencing methodology, the statistical analysis, etc.
- The manuscript does not present a specific section for conclusions, but it does have a detailed Results section and a Discussion section where the main question was addressed. A conclusion regarding which would be the best DNA isolation protocol is not possible, as it will depend on different factors, where the applicability of the method plays a predominant role, but the authors did provide their considerations and suggestions on what the best options would be, based on the study results.
- The references are appropriate and current. The manuscript contains 32 references, 66% of them from the last 5 years and 84% from the last 10 years. In addition, the style used in the references corresponds to that established by the journal.
- In the tables and figures of the manuscript and in the supplementary information provided by the authors, the scientific community interested in this field can find references that allow them to direct their research/studies.
Reviewer 4 Report
Comments and Suggestions for Authors
In this manuscript, the authors describe an extensive comparison of methods for DNA extraction from dried blood spots and the suitability of the DNA for next-generation sequencing. Overall, the manuscript is very well written and presents very thorough evaluation of method features and DNA quality. It would be helpful if the authors discussed other studies that compare different methods for DNA extraction from this specimen type (https://doi.org/10.3390/ijns6020036 - this is cited, but only in the methods – and https://doi.org/10.3390/pediatric17020030) and how those results compare to their own.
